# Dynamic supply-side multipliers in China's marine economy: A neural network-enhanced Ghosh model for sustainable development

Jian Jin◉, Mingqi Zhang◉ᴵᴰ◉*

School of Economics, Hebei University, Baoding, China

◉ These authors contributed equally to the work.
* zhangmingqi805@163.com

## Abstract

With the continuous growth of China's economy, marine economy plays an increasingly important role in the national economy. This study quantifies multiplier effects and supply-side dynamics in China's marine economy (2017–2023) to inform sustainable development strategies. Combining the Ghosh model, employment analysis, and structural path analysis (SPA), we enhance traditional input-output frameworks with LSTM neural networks to capture nonlinear sectoral interdependencies. Key results reveal marine tourism as the dominant contributor to value added (44.346%), gross output (48.87%). Marine fishery exhibits the highest direct employment coefficient (0.42681), while marine mining drives significant indirect job growth (coefficient: 0.35072) in related industries. Marine transportation ranks first in income multiplier (8.60929) and employment multiplier (3.0332), highlighting its pivotal role in household income. By innovatively integrating the Ghosh model with LSTM, this research overcomes static and linear limitations of conventional methods, providing policymakers with actionable insights for balanced sectoral development through optimized resource allocation and infrastructure investment.

## 1. Introduction

With the development and deepening of China's "Belt and Road Initiative" policy, maritime connectivity plays an increasingly important role in promoting economic and trade cooperation, and marine economy has also become an important part of China's economic growth [1]. China's marine economy has played a role in promoting production and development, increasing government fiscal and tax revenue, increasing employment, increasing economic income, and expanding currency recovery. Since the 21st century, China's marine economy obtained the fast development [2], from 2017 to 2023, successively put forward in the government work report "Promoting the Construction of Marine Economy Demonstration Area", "Strengthen Marine Economy", "Vigorously Develop the Blue Economy", "Develop Ocean Economy",

**Data availability statement:** All relevant data are within the manuscript and its Supporting Information files.

**Funding:** The author(s) received no specific funding for this work.

**Competing interests:** The authors have declared that no competing interests exist.

"Actively Expand Marine Economy Development Space" and "Develop the Marine Economy, Build a Marine Power ". The 20th report of the CPC has made a strategic deployment of "Develop the Marine Economy, Protect the Marine Ecological Environment, and Accelerate the Construction of a Marine Power" [3].

In the 21st century, with the increasingly mature theoretical research on marine economy, domestic scholars' research on industrial linkage has been increasing. For example, Xu and Chen [4] earlier analyzed the correlation analysis among the primary, secondary and tertiary industries in the marine industry and the correlation and influence analysis of the marine industry on other industries in the national economy. He and Wang [5] explained the development relationship between the marine industry and other industrial sectors according to the forward and backward correlation of the marine industry, which opened up ideas for the subsequent research on the correlation of the marine industry [6,7]. In 2010, studies on marine industrial effects showed a trend of diversification, and related studies on the coordinated development of land and sea economies began to emerge [8]. At this time, marine economic studies showed a scene of "A hundred flowers bloom and a hundred schools of thought argue". Scholars tried to use input-output method to analyze marine economic effects. For example, Wu, et al. [9] combined input-output and DEA model to measure the production efficiency of marine economy, and Zheng, et al. [10] used entropy method and input-output method to comprehensively evaluate the marine industry in the east sea area of China. In addition to the above research, the input-output analysis technology is commonly used in marine and land industrial economy [11,12], and the selection of leading industries [13].

Input-output analysis has been widely used in economic research in recent years, especially in the following specific fields, due to its powerful functions in describing industrial linkages, assessing economic impacts and resource and environmental flows:

1) Analysis of industrial association and contribution of marine economy

Xiang, et al. [14] used the input-output model, a dynamic analysis of the interrelationships and ripple effects among marine industries was conducted on the annual marine input-output tables of Zhejiang Province in China. Hong, et al. [15] constructed the input-output tables of the China's marine economy for the years 2002, 2007, 2012, 2017 and 2020, calculated the scale of marine industry development and its contribution to the national economy, and revealed the interrelationships and dynamic change trends between marine industries and land-based industries. Liang, et al. [16] used the stripping coefficient method to construct the input-output table of China's 11 coastal provinces' marine economy in 2017. Han, et al. [17] used the GRAM-DID model to evaluate the net effect of the establishment of the blue economic zone, and compiled the marine economy of Shandong Province input-output model to estimate the economic growth effects of other industries induced.

2) Economic impact assessment of extreme events and disasters

Wu, et al. [18] used the inoperability input-output model (IIM) and multiregion input-output model (MRIO) to simulate the economic consequences of nuclear

wastewater discharge in Japan and calculate the economic changes of each industry and country. Jin, et al. [19] toke Guangdong Province as the research object to evaluate the damage caused by storm surge disasters, determined the indirect economic losses of storm surge disasters between 2007–2017 by calculating the direct and indirect consumption coefficients. Liu, et al. [20] provided an assessment of the indirect economic losses caused by the sea ice disaster in Liaoning Province in China using an adaptive regional input-output (ARIO) model.

3) Environmental economic accounting and resource flows (carbon emissions and water footprints)

Zhao, et al. [21] integrated the China provincial $CO_2$ emission inventory data (1987–2017) to analyze trends and identify the main driving factors behind regional economic linkages and carbon emissions changes. Xu, et al. [22] explored the effects of sector aggregation on the embodied carbon emission of the residential consumptions of Beijing and Shanghai in 2012 and 2017 based on the city-centric global multi-region input-output (CCG-MRIO) model. Xu, et al. [23] used the physical value input-output table and the fish growth model, and calculated the output and input of China's aquatic products virtual water trade indirectly by fishery water coefficient and virtual water of feed crops. Zhu, et al. [24] proposed an inter-country input-output decomposition framework that can distinguish domestic firms and multinational enterprises and recalculated global value chain emissions. Li, et al. [25] toke China as the case, identified critical transmission sectors for food-water nexus at the provincial level, based on an environmentally-extended multiregional input-output model and the concept of betweenness. Li, et al. [26] identified critical transmission sectors and supply chain paths in China's virtual multi-regional grey water (VMGW) network, based on the betweenness concept, multi-regional input-output analysis, and structural path analysis.

Recent advancements in machine learning, particularly neural networks, have shown significant potential in economic modeling due to their ability to handle complex, nonlinear datasets. For instance, Chen [27] used the EA-ANN to integrate machine learning and IoT to optimize sustainable agriculture, enhancing resource management, enabling data-driven predictions for economic viability, and reducing environmental impacts through precision farming, while Liu [28] employed a two-hidden-layer MLP with data preprocessing to analyze digital economy-regional growth dynamics, achieving 55% accuracy and 45% F1 score, outperforming prior research in capturing non-linear impacts. These studies highlight the complementary role of neural networks in traditional economic models.

However, there are still significant gaps in the existing research:

1) The neglect of supply-side dynamics. Most studies focus on the demand side, lack in-depth discussion on the driving mechanism of the supply side in the marine economy, and it is difficult to reveal the "driving effect" of resource-intensive industries on the overall economy.

2) Limitations of static and linear assumptions. Traditional input-output models rely on static structures and linear relationships, which cannot fully capture the nonlinear dynamic associations caused by exogenous variables such as policy adjustments and environmental changes in the marine economy.

3) Insufficient quantification of the effects of China's marine economy. The existing literature rarely systematically quantifies the multiplier effects of various sectors of China's marine economy, and does not reveal the indirect economic contributions of key industries and their sustainability paths.

It can be seen from the research of the above scholars that the input-output method, as a relatively mature theory, has been widely used in the study of marine economy, but there are few studies on the economic effect of China's marine economy in 2017–2023. Based on this, the innovation points of this paper are as follows. Combining the neural network (LSTM) with the Ghosh model to overcome the limitations of traditional input-output models, such as their static nature and linear assumptions, when capturing the dynamic correlations in the marine economy. Adopting the Ghosh model to analyze the output effects of China's marine economy, this approach is particularly suited to China's marine economy,

where resource-intensive industries play a pivotal role in pushing economic activities. By quantifying the direct, indirect, and induced outputs generated by initial investments in marine sectors, the Ghosh model provides a robust framework to capture the ripple effects of supply-side policies on the broader economy. The employment model is used to calculate the number of jobs in related industries brought by China's marine economy and the percentage of the number of jobs in the total number of jobs in China. The structral path analysis (SPA) method is adopted, using input-output method to analyze the China marine economic multiplier effect, specific to the marine income multiplier effect, marine employment multiplier effect.

## 2. Data basis

### 2.1 Classification of input-output table of China's marine economy by sector

The input-output table of China's marine economy satisfies the basic form of input-output table [29], which is divided into intermediate input (intermediate use), final use and added value, namely quadrants I, II and III. In order to meet the requirements of row balance, column balance and total balance, this paper discusses the basic balance relationship of input-output table. This provides an important analysis basis for the input-output table of China's marine economy, and can deeply reveal the interdependence and mutual restriction among marine industries and between marine industries and land industries. The basic table of input-output of China's marine economy is shown in Fig 1.

The input-output table of China's marine economy contains a total of 28 sectors. This paper refers to the **Classification of Marine and Related Industries** (GB/T 20794−2021) issued by the National Marine Information Center and the **Classification of National Economic Industries** (GB/T 4754−2017) formulated by the Department of National Economic Accounting of the National Bureau of Statistics, and classifies 176 sub-categories according to relevant standards.

**Fig 1. Basic table of input-output table of China's marine economy.**

Through the analysis of China's competitive input-output table, 15 marine industry sectors are successfully extracted, and the other 13 marine related sectors are classified into the land sector, and finally the merging results of China's marine economic industry sectors and their corresponding relationships with China's input-output table are obtained (Due to the limited space, the results of the merger of China's marine economic industrial sectors and their corresponding relationship with China's input-output table are shown in S2 Table). The first quadrant of the table shows the data of intermediate input (intermediate use) of 28 sectors of China's marine economy, which is the core part of the input-output table. It fully reveals the interdependence and mutual restriction of technical and economic links between marine industrial sectors and between marine industrial sectors and land sectors. In this study, the horizontal axis represents the amount of marine industrial raw materials provided by sector $i$ to the relevant departments, and the vertical axis reflects the consumption of these raw materials by sector $j$. The second quadrant, as an extension of the first quadrant, shows the final use of each sector of the marine industry in terms of consumption, capital formation, and import and export. The third quadrant shows the added value of each sector, including worker's remuneration, net production tax, depreciation of fixed assets and operating surplus, so as to reflect the overall added value of each sector and its components [30].

## 2.2 Compilation of input-output table of China's marine economy

According to China's input-output accounting system, China's conducts input-output surveys and compiles input-output tables in every 2 and 7 years, and compiles extended input-output tables in every 0 and 5 years. To compile the input-output table of China's marine economy, it's necessary to investigate and count the data of various sectors of the national economy according to the actual data of the marine industry [31], calculate the stripping coefficient, form the stripping coefficient matrix, and complete the final compilation work through matrix operation. On the basis of the sectoral classification of the input-output table of the marine economy, the compilation steps are followed.

**2.2.1 Separating the data of each sector of China's marine economy by matrix.** The value-added rate in literature [32] is used to calculate the comprehensive index of industrial chain, and the stripping coefficient is used to distinguish the sea-land economic linkage. This paper mainly uses the statistical method to calculate the stripping coefficient $q$, and the steps are as follows:

$$\delta_j = \frac{VA_j}{Y_j} = \frac{V_j}{M_j} \tag{1}$$

$$q_j = \frac{M_j}{Y_j} = \frac{V_j}{VA_j} \tag{2}$$

where $V_{Aj}$ is the sum of the added value of each sector in the input-output table corresponding to the $j$th marine industry, $Y_j$ is the sum of the total output of each sector in the input-output table corresponding to the $j$th marine industry. $V_j$ is the added value of the $j$th marine industry, and $M_j$ is the total output value of the $j$th marine industry. $q_j$ is the stripping coefficient of the $j$th marine industry, and $\delta_j$ is the added value rate of the $j$th national economic industry. Based on the available characteristics of marine industry data and public data such as **China Statistical Yearbook** and **China Marine Economic Statistical Bulletin**, the stripping coefficients of each marine industry sector are calculated to determine the corresponding separation ratio [33].

Taking the stripping coefficient calculation of the marine fishery in 2020 as an example, the input-output table sectors corresponding to the marine fishery in 2020 in China are fishery products and agriculture, forestry, animal husbandry and fishery service products. From the input-output table of China in 2020, it can be obtained that the added value of these two sectors is 767,479.2323 million CNY and 336,533.9888 million CNY respectively, and the total output is 1,281,178.025 million CNY and 702,983.3185 million CNY, respectively. By summing up, the total added value of the corresponding

sectors of the marine fishery is 1,104,013.221 million CNY, and the total output is 1,984,161.343 million CNY. The added value rate of the marine fishery can be calculated as 0.5564. From the ***China Marine Economic Statistical Bulletin 2020***, it can be obtained that the added value of the marine fishery in 2020 is 471,200 million CNY. It is important to note that the ***China Marine Economy Statistical Bulletin 2020*** has no coastal beach planting industry, marine aquatic products processing industry, marine engineering equipment manufacturing industry, marine chemical industry, water desalination and comprehensive utilization of the related data. The data of these sectors need to be calculated according to the relevant data on the total added value and growth rate of marine industry in 2021 in the data table of gross marine product in ***China Marine Economy Statistical Bulletin 2022***. In 2020, the added value of the marine fishery is 471,200 million CNY, the added value rate of the marine fishery sector is 0.5564, and the total output of the marine fishery can be calculated as 846,852.9245 million CNY. Based on the above calculation, the total output of the marine fishery is 846,852.9245 million CNY, and the corresponding total output of the national input-output table of the marine fishery in 2020 is 1,984,161.343 million CNY. The stripping coefficient of the marine fishery sector is 0.42680648. The calculation method of divestment coefficient of other marine industry sectors is consistent with that of the marine fishery. Thus, the divestment coefficient matrix is constructed, and each sector of the marine economy is separated from each sector in China's competitive input-output table by means of the matrix divestment coefficient method [34].

**2.2.2 Adjusting the input-output table according to the GDP of the statistical yearbook.** After the economic census, the country has revised the GNP and its composition for the historical years, which has led to inconsistencies with the input-output table data. Therefore, it's necessary to adjust the input-output table data according to the GDP in the ***China Statistical Yearbook*** to ensure that the analysis of economic activity is more in line with the actual situation. Annual GDP figures from the ***China Statistical Yearbook*** as the added value of China's marine economic input-output table after the adjustment combined, for not covering marine economy industry sector in the Yearbook, calculating the added value and the ratio of input and output table corresponding to the added value of department, as the added value of the total after adjustment coefficient calculation. The structural proportion is determined according to the proportion of added value of each sector, and the four initial inputs, such as worker's remuneration, net production tax, depreciation of fixed assets and operating surplus, are adjusted accordingly [35]. The gross domestic product and its composition of the yearbook are taken as the total amount of each pair of expenditure adjusted in the second quadrant of the input-output table, and then the items are adjusted according to the structural proportion. Keeping the data in the first quadrant unchanged, the balance between input and output is used to recalculate the total input according to the adjusted added value. The total input series is used as the target total output series, and the balance relationship in the input-output table is used to calculate the total output according to the adjusted value of the second quadrant, and the difference between the calculated total output and the target total output (total input) is used as the other items.

**2.2.3 Construction of input-output table of China's marine economy in consecutive years.** Through the above operations, the input-output table of China's marine economy and industry can be separated from the original input-output table, and the input-output table of China's marine economy and industry in 2017, 2018 and 2020 can be obtained, including 15 marine economy and industry sectors, 13 non-marine economy and industry sectors, a total of 28 sectors. To meet the research needs, this paper still need interpolation input-output table in 2019, China's marine economy industry, and will be consecutive year China marine economy industry input-output table extended to 2023.

When the matrix transformation forecasting method (MTT) [36] is used to fill in the input-output table for the undisclosed years, it is necessary to calculate the three-part structure of the intermediate flow matrix in the missing years, the proportion of the final use, the added value structure and the final use structure. First of all, using linear interpolation method to calculate three parts structure in 2019, again with the help of moving average method forecast 2021–2023 three parts structure. According to ***China Statistical Yearbook 2023*** reported GDP calculation in 2019, 2021, 2022 and 2023, the added value and the final use of each component, on the basis of the middle traffic matrix proportion of final use further calculate the middle flow, eventually form 2017–2023 time series of China's marine economy industry input-output table.

## 2.3 Neural network stability and control in marine economic modeling

The integration of neural networks, particularly Long Short-Term Memory (LSTM) architectures, into the Ghosh input-output model enhances the predictive accuracy of marine economic dynamics by capturing nonlinear relationships and temporal dependencies in sectoral outputs. Ensuring neural network stability is critical for reliable long-term forecasting [37]. Stability in this context refers to the robustness of the network against input perturbations and its ability to avoid over-fitting. Techniques such as dropout regularization, batch normalization, and adaptive optimization algorithms (e.g., Adam) were employed to stabilize training, particularly given the high-dimensional and time-sensitive nature of input-output tables [38]. The LSTM's inherent gating mechanisms further mitigate vanishing gradient issues, preserving long-term dependencies in marine sector data (e.g., transportation and tourism trends), thereby maintaining consistent prediction performance across multi-year projections.

For neural network control, the framework leverages predictive outputs to inform dynamic policy adjustments [39]. By embedding feedback loops, the model continuously compares predicted sectoral outputs (e.g., employment multipliers, GDP contributions) with real-world economic indicators. Discrepancies trigger iterative retraining of the LSTM using updated data, optimizing parameters to align predictions with sustainable development goals [40]. Additionally, reinforcement learning strategies could be integrated to simulate policy interventions (e.g., resource allocation for offshore energy or tourism), enabling policymakers to evaluate the stability and economic impact of supply-side adjustments before implementation. This closed-loop control mechanism ensures adaptive governance of marine industries, balancing growth with ecological constraints while maintaining computational stability and interpretability.

The marine industry is significantly influenced by exogenous variables such as policies and environments, and the traditional Ghosh model is unable to quantify such dynamic effects. LSTM captures long-term temporal patterns through memory units and generates more accurate departmental output predictions. The input-output table of the marine economy covers 28 departments × 7 years of data, which is prone to overfitting. By using dropout regularization and batch normalization, the model maintains prediction stability under perturbations and ensures the reliability of multi-departmental collaborative analysis. This innovation integrates into supply-side policies as an assessment tool that is both accurate and robust.

## 3. Research on the output effect of China's marine economy

### 3.1 Input-output model

Input-output model is a kind of economic mathematical model, which is used to comprehensively analyze the quantitative dependence between input and output in economic activities [41]. This model especially emphasizes the analysis and investigation of the quantitative dependence between the production and consumption of products in various sectors of the national economy. In China's marine economy, due to less use of such models for scientific and accurate estimation and calculation, the quality of China's marine economy development in the past few years is not high enough. As China gradually proposes to enter the economic model of high-quality development, high-quality economic development should be paid more and more attention. Therefore, the application of input-output model in the field of the marine economy is conducive to improving the high-quality development of China's marine economy.

Literature [42] adopts multi-regional IO model (MRIO) and Ghosh model to simulate the driving effect of port investment on regional economy. The standard method for assessing inter-industry linkages is the traditional representation of the input-output relationship in the marine economy, as shown in Equation (3):

$$X = Z_i + F = AX + F \rightarrow X = (I-A)^{-1}F = LF \tag{3}$$

where matrix $X = [x_i]_{n \times 1}$ represents the sum of transaction flows between activity sectors and total output, $Z = [z_{ij}]_{n \times n}$ is the intermediate use matrix, matrix $I$ is the identity matrix, vector $X$ is the total output of the marine economy, vector $F = [f_i]_{n \times 1}$

represents the final demand portion of the total output of the marine economy sold, $L=[l_{ij}]_{n\times n}=(I-A)^{-1}$ is the inverse Leontief matrix and represents the sum of the direct and indirect output of external final demand in each unit of the $i$th marine sector, and $A$ is the input coefficient matrix, defined as follows:

$$A=[a_{ij}]_{n\times n}=[\frac{z_{ij}}{x_j}]_{n\times n}$$

(4)

where $z_{ij}$ represents the median input demand of the $i$th marine sector and the $j$th marine supply sector, and $x_j$ represents the final output of the $i$th marine sector.

### 3.2 Ghosh model

So far, as a result of the limitation of methods and data, especially from the perspective of the supply side, the impact of China's marine economy on output has received little attention. The existing input-output models to some extent ignore the interaction between the marine industry and other industrial sectors, which may underestimate the contribution of the marine economy to the overall economic output. As a result, the input-output model combined with Ghosh model, establishing a supply-side framework for analysing the ocean economic relationship with China's overall economic framework is of great significance. This helps to systematically understand the drivers and linkages of marine resources-related industrial output, thus intuitively revealing the important role of marine economy in economic growth and structural adjustment.

Book [43] identifies the radiation ability of key industries to upstream and downstream supply chains through Ghosh supply-driven model. In the traditional input-output model, the final demand is completely determined externally [44], and the commonly used expression is shown in Equation (5).

$$X=Z_i+F=AX+F=(I-A)^{-1}F=LF$$

(5)

where $X$ is the total output vector, $Z$ is the intermediate use matrix, $F$ is the final demand vector, $A$ is the input coefficient matrix, $i$ is the identity column vector, $I$ is the identity matrix, and $L$ is the Leontief inverse matrix.

Using a semi-closed input-output model to endogenize part of the final demand allows the inter-sectoral interaction and feedback mechanism within the model to be more complex and real [45], which can more accurately capture the interaction between the marine industry and other economic sectors and reveal the real contribution of the marine economy to the overall economic output, thus providing a scientific basis for the formulation of more effective marine economic policies. The calculation process of semi-closed model used is shown in Equation (6).

$$\begin{bmatrix} X \\ x_{n+1} \end{bmatrix} = \begin{bmatrix} A & h_C \\ h_R & 0 \end{bmatrix} \begin{bmatrix} X \\ x_{n+1} \end{bmatrix} + \begin{bmatrix} g \\ 0 \end{bmatrix}$$

(6)

where $x_{n+1}=\sum_{j=1}^{n} w_j$ is the output of China's marine economy, $W=[w_j]_{1\times n}$ is defined as the row vector of the labor compensation of employees, $h_C=[a_{i,n+1}]_{n\times 1}$ is the column vector of marine consumption expenditure coefficients, $a_{i,n+1}=c_i/x_{n+1}$, $C=[c_i]_{n+1}$ is defined as the column vector of final consumption, $h_R=[a_{n+1,i}]_{1\times n}$ is the row vector of labor input coefficients, $a_{n+1,j}=w_j/x_j$, and $g$ is the surplus value of external demand.

The solutions of Equations (7) are given by

$$\begin{bmatrix} X \\ x_{n+1} \end{bmatrix} = \begin{bmatrix} I-A & -h_C \\ -h_R & 1 \end{bmatrix}^{-1} \begin{bmatrix} g \\ 0 \end{bmatrix}$$

(7)

The Ghosh inverse matrix can be defined in the Ghosh model based on the semi-closed input-output model [46]:

$$G^* = (I - B^*)^{-1} = \begin{bmatrix} I - B & -h_C{}' \\ -h_R{}' & 1 \end{bmatrix}$$

<div align="right">(8)</div>

where $B$ is the direct output coefficient matrix, $h_C{}'$ is the column vector of consumption coefficients, $h_R{}'$ is the row vector of labour input coefficients.

Total output is determined by economic added value, and the expression of output can be expressed by Equation (9).

$$X^* = V^* (I - B^*)^{-1}$$

<div align="right">(9)</div>

where $V^*$ is the row vector of input-output added value, and $B^*$ is the matrix of complete supply coefficients.

By further extending and generalizing the Ghosh model, Equation (10) can be obtained.

$$E = pwNG^* \hat{\varepsilon}$$

<div align="right">(10)</div>

where $p$ is a scalar population coefficient, $w$ is the scalar of per capita primary input volume, $N$ is the row vector of supply-side economic structure, $E$ is the column vector of marine industry sector input, and $G^* \hat{\varepsilon}$ is the impact of marine industry sector input on the total output of the whole supply chain from the perspective of primary input.

### 3.3 Differences between the Ghosh supply-push model and Leontief demand-pull model

The Ghosh supply-push model and Leontief demand-pull model represent two distinct analytical frameworks in input-output economics, each emphasizing opposing drivers of economic activity [47]. The Ghosh model focuses on the supply side, examining how primary inputs (e.g., labor, capital, raw materials) propagate forward through production chains to generate sectoral output. Its mathematical formulation, $X = (I-B)^{-1}V$, employs a direct output coefficient matrix $B$ to quantify forward linkages, where $V$ represents primary inputs [48]. This model is particularly suited for analyzing resource-intensive industries (e.g., offshore oil, marine mining) and evaluating supply-side policies such as infrastructure investments, as it captures how initial resource allocations push downstream industries to expand production.

In contrast, the Leontief model centers on the demand side, exploring how final demand (e.g., consumption, exports) propagates backward through supply chains to stimulate upstream production [49]. Its equation, $X = (I-A)^{-1}F$, utilizes a direct input coefficient matrix $A$ and final demand vector $F$ to measure backward linkages. This framework is ideal for assessing demand-driven impacts, such as tourism growth or export surges, and policies like consumer incentives, as it quantifies how demand shocks ripple through upstream sectors. While Ghosh emphasizes supply-driven economic "pushes" (e.g., resource availability), Leontief highlights demand-driven "pulls" (e.g., market expansion). Together, these models offer complementary insights: Ghosh informs resource allocation and infrastructure planning, while Leontief guides market strategies and demand-side interventions. Integrating both enables policymakers to balance supply-chain robustness with demand-driven growth for sustainable economic development.

The Ghosh model excels in quantifying supply-driven dynamics, making it particularly suitable for analyzing resource-intensive marine sectors (e.g., offshore energy, marine mining) where initial resource allocations or infrastructure investments act as primary economic drivers [50]. Its forward-linkage mechanism effectively traces how primary inputs (e.g., capital injections into port infrastructure) propagate downstream, revealing indirect output and employment effects in related industries (e.g., equipment manufacturing). However, its applicability is constrained by three key limitations in marine contexts. First, its static and linear assumptions struggle to capture nonlinear disruptions common in marine systems, such as policy shifts (e.g., fishing moratoriums) or environmental shocks (e.g., oil spills), which alter sectoral interdependencies. Second, it overlooks demand-side fluctuations (e.g., tourism seasonality), potentially overestimating the sustainability of supply expansions in demand-volatile sectors like marine aquaculture. Third, its heavy reliance on

accurate primary input data poses challenges for emerging marine industries (e.g., marine renewables), where statistical granularity is often inadequate. Consequently, while the Ghosh model is indispensable for supply-side policy planning, its standalone use risks misjudging systemic vulnerabilities in dynamic marine economies.

The Leontief model offers precision in demand-side analysis, ideal for marine sectors dominated by final consumption or export dependencies (e.g., marine tourism, seafood exports). Its backward-linkage framework quantifies how demand shocks (e.g., a surge in cruise tourism) ripple upstream, informing policies targeting market expansion or consumer incentives. Yet, its limitations are pronounced in marine resource economies [51]. Primarily, its neglect of supply-side rigidities—such as resource scarcity (e.g., depleted fisheries) or infrastructure bottlenecks (e.g., port capacity constraints)—may overestimate the efficacy of demand-stimulus policies. For instance, promoting marine tourism without concurrent port investments could inflate demand but strain existing logistics, dampening actual output growth. Additionally, its static structure fails to adapt to exogenous marine-specific variables, such as international trade disputes disrupting seafood demand or climate-induced shifts in shipping routes. These constraints are especially acute in supply-constrained marine sectors (e.g., marine mining), where ignoring resource availability may lead to unrealistic growth projections. Thus, while the Leontief model is powerful for demand-driven scenarios, its blind spots necessitate complementary frameworks to address marine-specific supply-demand imbalances.

## 3.4 Integration of neural networks

To enhance the Ghosh model's predictive capability, we employ a long short-term memory (LSTM) neural network to forecast key parameters in the input-output table (e.g., sectoral output, intermediate demand). The LSTM architecture is chosen for its proficiency in handling time-series data and capturing long-term dependencies. The workflow is as follows:

Data preprocessing: historical input-output tables (2017–2023) are normalized and split into training (2017–2021) and validation (2022–2023) sets.

Model training: the LSTM network (2 hidden layers, 64 units each) is trained to predict sectoral output based on lagged values of GDP, employment, and final demand.

Validation: model accuracy is evaluated using mean absolute percentage error (MAPE), achieving a score of 3.2%, outperforming linear interpolation methods.

The neural network forecasts reveal that the marine transportation and tourism sectors exhibit nonlinear growth patterns, driven by policy shifts and technological adoption. These insights refine the Ghosh model's supply-side projections and inform balanced sectoral development strategies. By using the above methods, it is possible to break through the traditional static input-output framework and construct a dual-engine analysis system of "supply-side drive + machine learning dynamic calibration". The output of the neural network is transformed into policy intervention parameters, achieving a sustainable governance path of "prediction - optimization - feedback".

## 3.5 Analysis of the output effect of marine economy

The added value data of China's marine industry is shown in Fig 2. It can be seen that the added value of worker's remuneration of the marine tourism is the largest, but the proportion of its net production tax is equal to that of marine transportation industry. In addition, the ratio of net production tax to worker's remuneration of the offshore oil and gas industry is higher. The proportion of depreciation of fixed assets is relatively high in the marine tourism and the marine transportation industry respectively. In terms of operating surplus, the marine transportation industry accounts for the highest proportion.

Through the above model analysis, the industrial added value matrix of China's main marine industrial sectors can be further obtained as shown in Table 1, and the industrial added value of different industries can be calculated. The added value of marine industry departments directly refers to the department in the process of production, through the

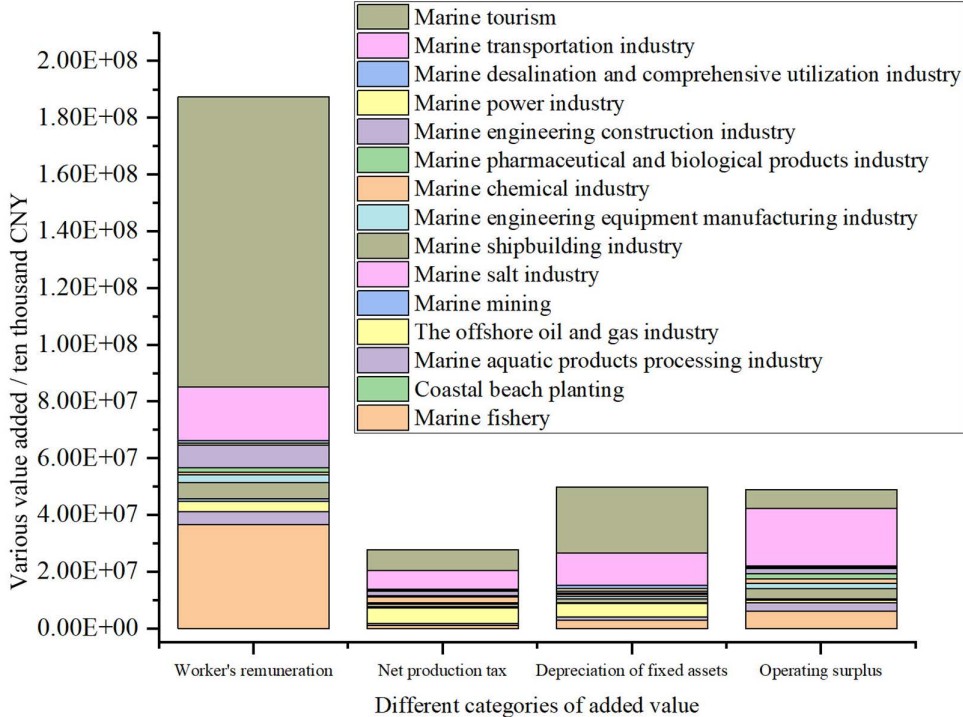

**Fig 2. Analysis of the added value of different marine industry categories.**

**Table 1. Industrial added value of China's marine industry sector in 2023.**

| Code | Name of Marine Industry | Industry Added Value (Million CNY) |
|---|---|---|
| 01 | Marine fishery | 471,199.9968 |
| 02 | Coastal beach planting | 203.6666231 |
| 03 | Marine aquatic products processing industry | 90,457.26 |
| 04 | The offshore oil and gas industry | 149,400 |
| 05 | Marine mining | 18,999.99036 |
| 06 | Marine salt industry | 3,299.99822 |
| 07 | Marine shipbuilding industry | 114,760.088 |
| 08 | Marine engineering equipment manufacturing industry | 57,330.26077 |
| 09 | Marine chemical industry | 53,199.99245 |
| 10 | Marine pharmaceutical and bioproducts industry | 45,099.99216 |
| 11 | Marine engineering construction industry | 119,000.0021 |
| 12 | Marine power industry | 23,699.99416 |
| 13 | Marine desalination and comprehensive utilization industry | 29,668.25965 |
| 14 | Marine transportation industry | 571,100.0457 |
| 15 | Marine tourism | 1,392,400.013 |

investment of labor, capital and technology, create new value directly. Analysis of different industrial sectors added value contributes to an understanding of the industry's contribution to the economy, development level and competitiveness, to identify growth point and the weak link, optimize resource allocation, promote industrial structure adjustment and

economic sustainable development. This is of great significance for governments to formulate industrial policies, enterprises to make strategic decisions, and academic research.

The industrial added value of the marine tourism reaches 1,392,400.013 million CNY, accounting for 44.346%, which is the largest and most prominent sector in the marine economy. Its labor remuneration accounts for the highest proportion, reflecting its high reliance on labor. Its net production tax contribution is outstanding, comparable to that of the marine transportation industry, indicating the significant boost to its tax revenue from policy support. Its fixed asset depreciation rate is relatively high, due to the large investment in infrastructure, and its operating surplus ratio is lower than that of the transportation industry, indicating higher operational costs.

The added value of marine fishery is 471,199.9968 million CNY, accounting for 15.01%. As a traditional pillar industry, it ranks third in terms of added value scale. It mainly relies on the production of primary products. The future development directions include: promoting intelligent breeding and cold chain logistics, developing high value-added products (such as functional foods), and deepening the industrial chain. The added value of the marine aquatic products processing industry is 90,457.26 million CNY, accounting for 2.881%. Although it is relatively small in scale, as an extension of the fishing industry, it is a key link for enhancing the added value of marine fisheries. Currently, its deep processing capabilities are weak, resulting in limited contribution to the added value. The potential areas include developing advanced deep processing technologies and connecting with the health food market.

The added value of the offshore oil and gas industry is 149,400 million CNY, accounting for 4.758%. As a core sector of energy security, it has a stable contribution to the added value, but it is highly technology-intensive and relies on advanced equipment and professional talents. Its net production tax accounts for a high proportion of the added value, reflecting its significant contribution to fiscal revenue. Future challenges include the need to balance resource development and ecological protection, and accelerate the transition to clean energy. The added value of the marine mining is 18,999.99036 million CNY, accounting for 0.605%. It has the smallest scale but stands out in terms of strategic resource value. Its added value rate is low, mainly due to the open and crude mining mode and technical bottlenecks. The future transformation directions include exploring green mining technologies, reducing environmental costs, and enhancing the added value of resources.

The added value of the marine shipbuilding industry was 114,760.088 million CNY, accounting for 3.655%. Its total output ranked third, but the added value rate was relatively low, indicating overcapacity at the mid-to-low end and insufficient proportion of high-end ships. The proportion of operating surplus was high, indicating strong profitability. The potential breakthrough points include upgrading to high-tech ships and increasing the global market share. The added value of the marine engineering equipment manufacturing industry is 57,330.26077 million CNY, accounting for 1.826%. It is a technology-intensive sector that provides key equipment for oil and gas exploration and offshore wind power. The scale of added value is medium-sized, but the pull effect of the industrial chain is strong. The following policies are needed: strengthening research and development subsidies and breaking through the technical barriers of core components.

The added value of the marine chemical industry is 53,199.99245 million CNY, accounting for 1.694%. It has a high proportion of basic chemicals and relatively low added value, and the high-end fields have not yet achieved large-scale development. The proportion of fixed asset depreciation is relatively high, reflecting the urgent need for equipment renewal. The high value-added directions include developing marine biobased materials and environmentally friendly marine anti-corrosion coatings. The added value of marine pharmaceutical and bioproducts industry is 45,099.99216 million CNY, accounting for 1.436%. It is a strategic emerging industry with significant high value-added characteristics. However, it is currently small in scale and is limited by long research and development cycles and low technology conversion rates. The core bottlenecks include strengthening cooperation among industry, academia and research institutions, and accelerating the industrialization of marine genetic resources.

The added value of the marine engineering construction industry is 119,000.0021 million CNY, accounting for 3.79%. It is a core sector that supports offshore wind power, cross-sea bridges, and port infrastructure. Its added value scale ranks

first among emerging sectors. The operating surplus proportion is relatively low, indicating high project costs and limited profit margins. Future growth points include focusing on green engineering and intelligent construction technologies. The added value of the marine power industry is 23,699.99416 million CNY, accounting for 0.755%. It is a representative of the low-carbon industry, mainly focusing on offshore wind power. The added value is small but has great growth potential. The high initial investment leads to a low added value rate, and it is necessary to adopt energy storage technologies to reduce costs and improve efficiency. The key policy lies in improving the electricity price subsidy mechanism and promoting the commercialization of offshore wind power technology.

The added value of the marine desalination and comprehensive utilization industry is 29,668.25965 million CNY, accounting for 0.945%. It is a strategic industry for solving water shortages in coastal areas, but its industrialization level is low. The added value contribution is limited, mainly due to insufficient large-scale application. The breakthrough paths include policy-driven large-scale demonstration projects and reducing energy consumption costs. The added value of the marine transportation industry is 571,100.0457 million CNY, accounting for 18.189%. It is the second largest pillar sector, with its added value scale approaching half of that of the marine tourism industry. It mainly comes from port logistics and shipping services. It plays an irreplaceable driving role in the overall economy. The future transformation requirements include developing green shipping and responding to the international carbon tax mechanism.

The added value of the marine salt industry is 3,299.99822 million CNY, accounting for 0.105%. The scale of its traditional industries has shrunk and its contribution to added value is weak. It relies on the traditional salt-making process with low added value, but as a basic raw material for chemicals, its supply chain value is irreplaceable. The future upgrade paths include promoting the integration of salt fields with tourism and culture, developing high-purity industrial salt and health salt products.The added value of the coastal beach planting is 203.6666231 million CNY, accounting for 0.0065%. Its value-added rate is extremely low. Due to its reliance on traditional salt-making and manual planting methods, which are of a simplistic and extensive nature, there is insufficient investment in technology. Its total output contribution is weak, accounting for a negligible proportion in the marine economy, but it has ecological value (such as the protection of mangroves).

The total output data of China's 15 marine industry sectors are shown in Fig 3 and Table 2. In the input-output table of the marine economy, total output usually refers to the total value of all goods and services produced by marine industries and their related industries within a certain period. This includes direct output, the value generated by the direct economic activities of the core marine industries. Indirect output: the economic activities of other non-marine sectors driven by the upstream and downstream connections within the industrial chain. And induced output, the output of other industries driven by the consumption of marine industry practitioners.

The total output value of marine tourism is 4,057,181.7563 million CNY, accounting for 48.87%, making it the absolute pillar industry of the marine economy. Its significant contribution mainly stems from national policy support and the rapid growth of domestic tourism demand. The total output value of marine transportation industry is 983,594.853 million CNY, accounting for 11.85%, ranking second. Its development benefits from the growth of international trade and the expansion of port logistics. The total output value of marine fishery is 827,736.05 million CNY, accounting for 9.97%. As an established industry, it plays a crucial role in ensuring food safety and employment. These three types of industries are the leading industries within the marine sector.

The total output value of marine engineering construction industry was 533,654.043 million CNY, accounting for 6.43%. The total output value of marine shipbuilding industry was 493,206.5423 million CNY, accounting for 5.94%. These two sectors play a crucial role in infrastructure construction such as ports and cross-sea bridges, promoting the coordinated development of the industrial chain. The total output value of the marine chemical industry was 299,945.378 million CNY, accounting for 3.61%. The total output value of the marine engineering equipment manufacturing industry was 275,473.643 million CNY, accounting for 3.32%. These two industries provide technical and equipment support for marine resource development, and have a high added value. These four types of industries are the important supporting industries for the marine industry.

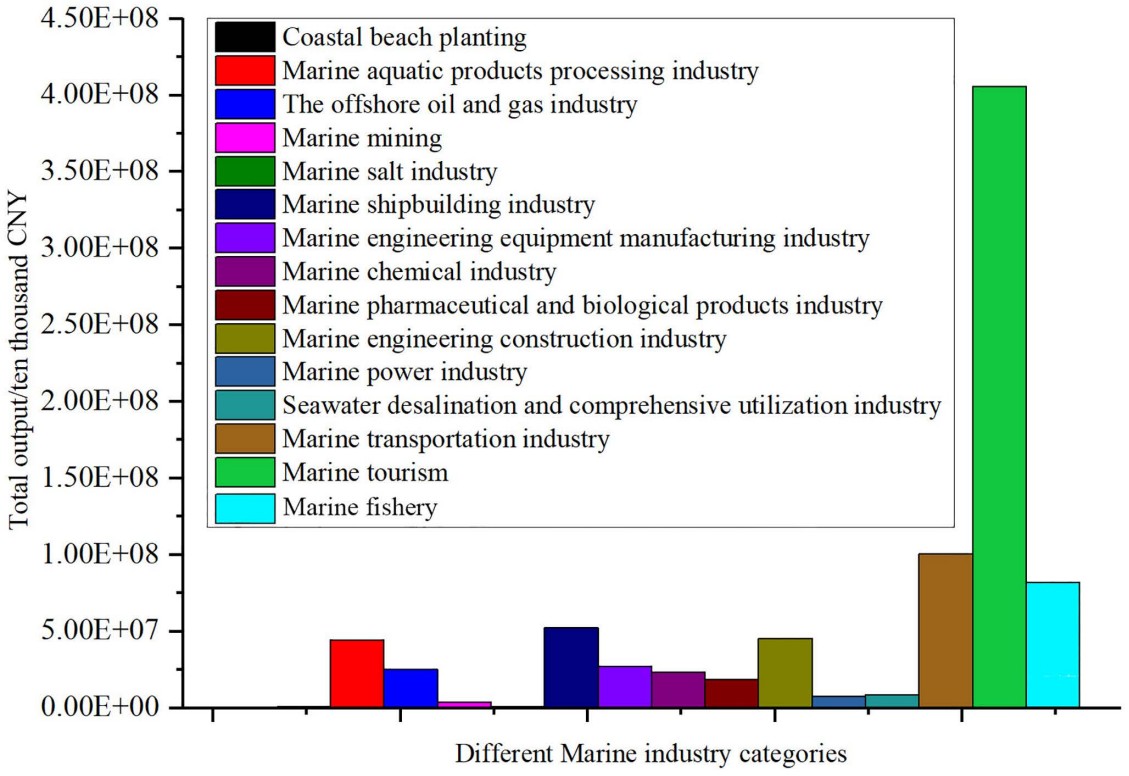

**Fig 3. Analysis of the total output of different marine industry categories.**

**Table 2. Industrial total output of China's marine industry sector in 2023.**

| Code | Name of Marine Industry | Industry Total Output (Million CNY) |
|---|---|---|
| 01 | Marine fishery | 827,736.05 |
| 02 | Coastal beach planting | 4,989.067 |
| 03 | Marine aquatic products processing industry | 391,770.66 |
| 04 | The offshore oil and gas industry | 103,909.75 |
| 05 | Marine mining | 15,543.7518 |
| 06 | Marine salt industry | 7,537.344 |
| 07 | Marine shipbuilding industry | 493,206.5423 |
| 08 | Marine engineering equipment manufacturing industry | 275,473.643 |
| 09 | Marine chemical industry | 299,945.378 |
| 10 | Marine pharmaceutical and bioproducts industry | 179,406.054 |
| 11 | Marine engineering construction industry | 533,654.043 |
| 12 | Marine power industry | 66,323.8537 |
| 13 | Marine desalination and comprehensive utilization industry | 61,567.07 |
| 14 | Marine transportation industry | 983,594.853 |
| 15 | Marine tourism | 4,057,181.7563 |

The total output value of the marine pharmaceutical and bioproducts industry was 179,406.054 million CNY, accounting for 2.16%. As a strategic emerging industry, it has a remarkable high value-added characteristic. However, due to the long research and development cycle and low technology conversion rate, its scale has not been fully unleashed. The

total output value of the marine power industry is 66,323.8537 million CNY, accounting for 0.80%. The total output value of seawater desalination and comprehensive utilization industry is 61,567.07 million CNY, accounting for 0.74%. These two industries have great potential under the background of green development and sustainable utilization of resources, but they need policy support to break through the bottleneck of large-scale operation. These three types of industries are the emerging and potential industries within the marine sector.

The total output value of marine mining is 15,543.7518 million CNY, accounting for 0.19%. The total output value of marine salt industry is 7,537.344 million CNY, accounting for 0.09%. The total output value of coastal beach planting is 4,989.067 million CNY, accounting for 0.06%. These three industries are peripheral industries in the marine sector, with extremely low contribution to the total output value. They need to undergo transformation and upgrading or integrate with tourism, culture and other industries to enhance their value.

## 4. Research on the employment effect of China's marine economy

### 4.1 Employment model

Ghosh model combining the Leontief inverse matrix of the $j$th column in the elements, to meet the demand of different products or services by the department of marine economy, the products of the different departments can be classified management [52]. By comparing the differences between Leontief demand model and Ghosh supply model, literature [53] proposed the complementarity of the two in the Marine industrial chain. The model of relationship between them as shown in Equation (11). This model reflects the indirect or direct effects of changes in final demand among different sectors of the marine economy on the total output of the marine economy.

$$m(o)_j = \sum_{i=1}^{n} l_{ij} \tag{11}$$

where $l_{ij}$ is the economic demand of different marine sectors. In order to estimate the marine economic induction effect, the model as above is often extended so that the marine industry final consumption expenditure becomes endogenous, and the capacity generated by this aggregate is used as an input in the production sector. In this model, input-output closure with respect to industry will generate an expanded matrix of technical coefficients and an expanded Leontief inverse matrix, both of dimension $n+1$. The elements of this inverse matrix include total impacts (direct, indirect, induced, etc.), and the sum of the nth elements in each column of the inverse matrix will represent the individual impacts of the total output multiplier on the nth marine industry sectors. In this way, the revised total output multiplier can be further obtained, which is calculated as follows:

$$\bar{m}[o(t)]_j = \sum_{i=1}^{n} \bar{l}_{ij} \tag{12}$$

The added value or income multipliers of economic industries in different sub-sectors can be further estimated. The calculation process and model can be referred to the following equation:

$$m(r)_j = \sum_{i=1}^{n} v_{ci} l_{ij} \tag{13}$$

$$\bar{m}[(r)]_j = \sum_{i=1}^{n} v_{ci} \bar{l}_{ij} \tag{14}$$

   

where the $v_{ci}$ represents the ability of different marine industry sectors to generate added value per unit of product or service.

Similarly, the use of the $L$ inverse matrix can be used to estimate the employment multiplier by obtaining employment demand through activities. This employment multiplier reflects the direct, indirect and total employment effects of increased final demand in different marine industry sectors, respectively.

$$m(e)_j = \sum_{i=1}^{n} e_{ci} l_{ij}$$

(15)

$$\bar{m}[e(t)]_j = \sum_{i=1}^{n} e_{ci} \bar{l}_{ij}$$

(16)

where $e_{ci}$ is the technical coefficient of employment, which represents the number of employed people required to produce a unit of output in the $i$th marine industry sector. From this, it is easy to obtain the total employment demand, and the sum of all employment demands indicates the total employment generated in the marine economy due to the increase in production in the different marine industry sectors.

## 4.2 Analysis of marine economic employment in China

Direct employment coefficient is an important indicator to measure the ability of an industry to directly create employment opportunities. A high direct employment coefficient means that the industry contributes more to employment. Through the analysis of the direct employment coefficient of each industry, we can better understand the contribution of each marine economic industry in terms of employment, and then provide a basis for policy making. Fig 4 shows the distribution of direct employment coefficients of different industries in China's marine economy in 2023.

The direct employment coefficient of the marine fishery is 0.42681, which is the highest among all industries, indicating that its contribution to employment is the most significant. Marine fishery in China's coastal provinces is dominated by capital and labor input, of which labor input accounts for a significantly higher proportion than other marine industries, and China's "blue growth" strategy emphasizes the construction of marine pasture and the development of recreational fishery to create more jobs through the extension of the industrial chain. The coefficients of the marine aquatic products processing and the marine tourism are 0.23524 and 0.26681, respectively, indicating that these industries are also creating a large number of employment opportunities. The high coefficients for these industries reflect that they require a large number of workers to carry out activities such as fishing, breeding, processing and services. According to a survey conducted by Zhejiang Ocean University, tourism accounts for 13.56 percent of the employment flow of local graduates in Zhoushan, second only to manufacturing [54]. Zhoushan promotes international activities such as "Island Tourism Conference" through the free trade zone policy, integrates land and sea resources in combination with the strategy of "promoting sea by land", and further expands the scale of tourism employment. Guizhou Province has promoted the upgrading of employment structure by developing primary and deep processing of aquatic products [55]. For example, the establishment of primary processing facilities reduces wastage and creates jobs; Develop product lines such as prepared dishes and surimi to extend the industrial chain and absorb labor. The policy requires the construction of a whole-process traceability system from breeding to processing, which further strengthens the employment stability of processing links.

In contrast, the coefficients for direct employment in the coastal beach planting industry and the marine mining are very low, at $3.8\text{E}^{-5}$ and 0.00816, respectively, indicating that these industries have a limited contribution to job creation. The coastal beach planting industry may be due to its small scale and low technology intensity, while the marine mining may be due to its high degree of automation and relatively low labor demand. The promotion of mechanized farming mode has

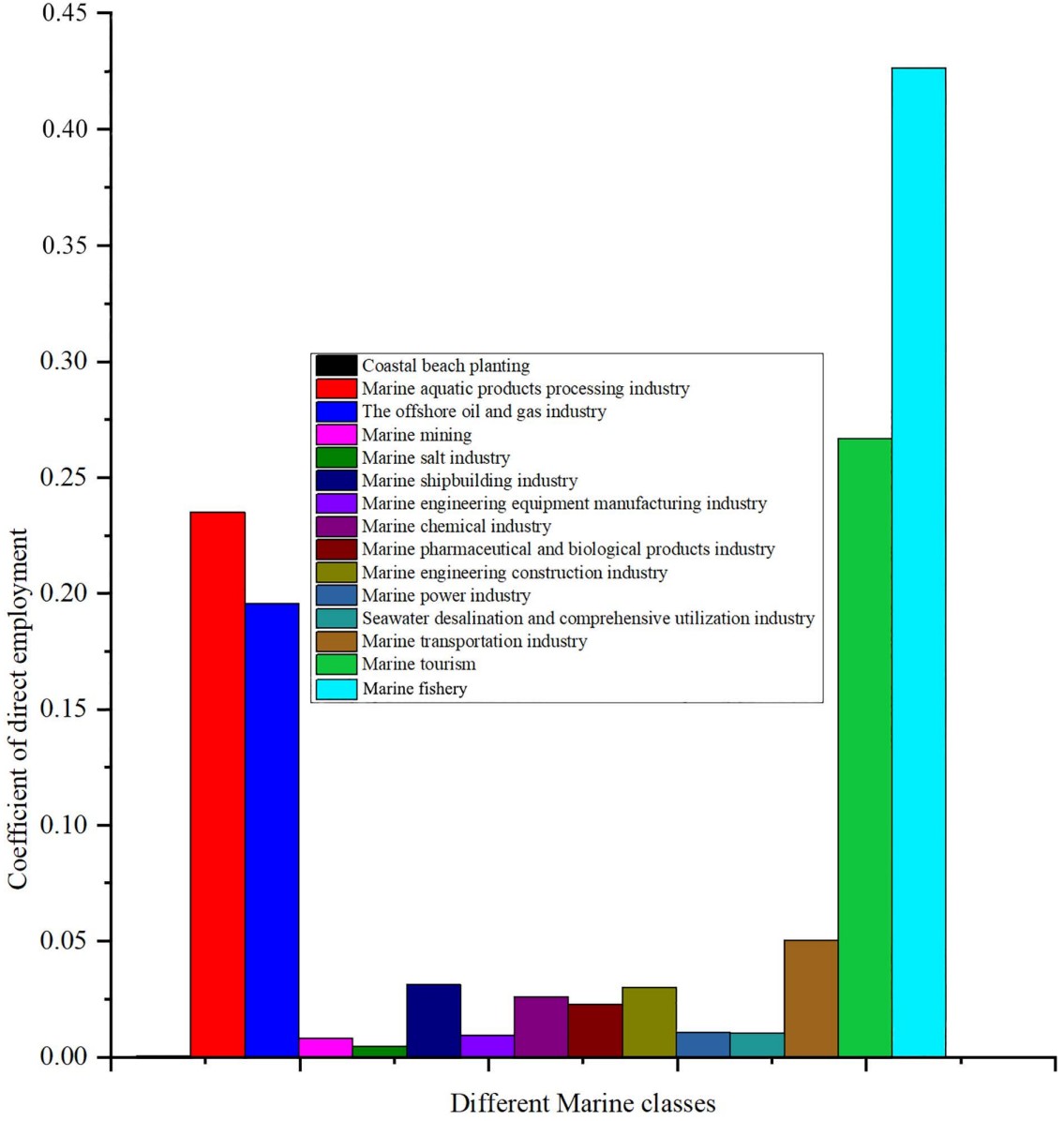

**Fig 4. Coefficients of direct employment for different marine industry classes.**

significantly reduced the dependence of traditional coastal beach planting on labor. For example, China's shellfish farming mechanization platform system integrates real-time monitoring, intelligent farming, automated harvesting and sorting technologies, increasing production efficiency by more than 30 times while significantly reducing labor requirements. In addition, in the production function of mining industry, the capital elasticity is much higher than the labor elasticity, and the marginal effect of increasing investment on employment growth is decreasing.

Other industries such as the offshore oil and gas industry, the marine chemical industry and the marine shipbuilding industry of direct employment coefficient is 0.1959, 0.02615 and 0.03136 respectively, shown that the contribution of these industries in terms of employment is moderate. Despite the higher coefficient of the offshore oil and gas industry,

its employment contribution is still less than that of the marine fishery and the marine tourism compared to the traditional manufacturing industry. The coefficients of the marine chemical industry and the marine shipbuilding industry are slightly lower, reflecting their higher technology intensity. The widespread use of automated production technology in the offshore oil and gas industry is the primary factor in reducing the need for direct labor. Remote monitoring systems are widely used on modern offshore platforms, with the number of resident personnel on a single platform dropping from an average of 60 in the 1980s to around 20 today, and the use of subsea production systems has even made some fields completely unmanned. The production process characteristics of the Marine chemical industry fundamentally determine the low employment coefficient. According to the Korea Marine Environmental Industry Research data, between 2010 and 2013, the number of jobs created in the Marine chemical industry per billion won of investment fell from 10.17 to 9.18, a decline of 9.7%, mainly due to the popularization of automated control systems [56].

The marine engineering construction industry, the marine power industry, and the marine transportation industry of direct employment coefficient is 0.03024, 0.01085 and 0.05064, respectively. Due to the large number of manpower needed for construction and maintenance, the employment coefficient of the marine engineering construction industry is relatively high. The low coefficients of the marine power industry and the marine desalination and comprehensive utilization industry reflect that these industries are more automated and technology-intensive and have less demand for direct labor. The capital intensive nature of large-scale projects in the offshore engineering construction industry significantly reduces the labor demand per unit of output. Taking the Hong Kong-Zhuhai-Macao Bridge as an example, the total investment of this world-class cross-sea channel project is as high as 126.9 billion yuan, creating a number of world records. However, the average annual direct employment during the construction period is only about 13,000, and the employment coefficient is about 0.01 people/ 10,000 yuan of investment, far lower than general civil engineering. The choice of policy instruments in the Marine power industry generates unexpected distortions in the employment coefficient. China's green power certificate trading policy, implemented since 2017, essentially encourages enterprises to choose large-scale and offshore projects with lower unit investment employment coefficient [57]. The downward trend in the employment coefficient of shipping enterprises in the Marine transportation industry is the most obvious. Global container shipping data show that between 2000 and 2020, direct employment per million ton-km transported fell 52% from 0.25 to 0.12 [58].

There are significant differences in the contribution of various sectors of the marine economy in terms of employment. The marine fishery and the marine tourism have made outstanding contributions to employment creation, while the marine mining and the coastal beach planting industry have less direct contribution to employment. These differences reflect the different characteristics of each industry in terms of labor demand, technology intensity and production scale.

The indirect employment coefficient is an important measure of an industry's ability to drive employment through its supply chain and related industries. A high indirect employment coefficient means that the industry contributes more to employment through the upstream and downstream industrial chain. For China's marine economy, the level of indirect employment coefficient directly affects the employment level of related industries and the whole economic system. Through the analysis of the indirect employment coefficient of each marine economy industry, we can better understand the contribution of these industries in driving the employment of other industries, which provides an important reference for optimizing the industrial structure and promoting employment. Fig 5 shows the distribution of indirect employment coefficients of different industries in China's marine economy in 2023.

In marine economy industries in China, the marine mining's indirect employment coefficient is 0.35072, shows its employment in other related industry has a significant impact. This may be due to the marine mining needs a lot of equipment, technical support and services, the demand for equipment manufacturing, transportation, maintenance and other related industry employment. In contrast, the indirect employment coefficients of the marine fishery and the marine shipbuilding industry are 0.01789 and 0.01736 respectively, which are not as significant as the marine mining, but still show their driving effect on the employment of the upstream and downstream industrial chains. The marine shipbuilding

 

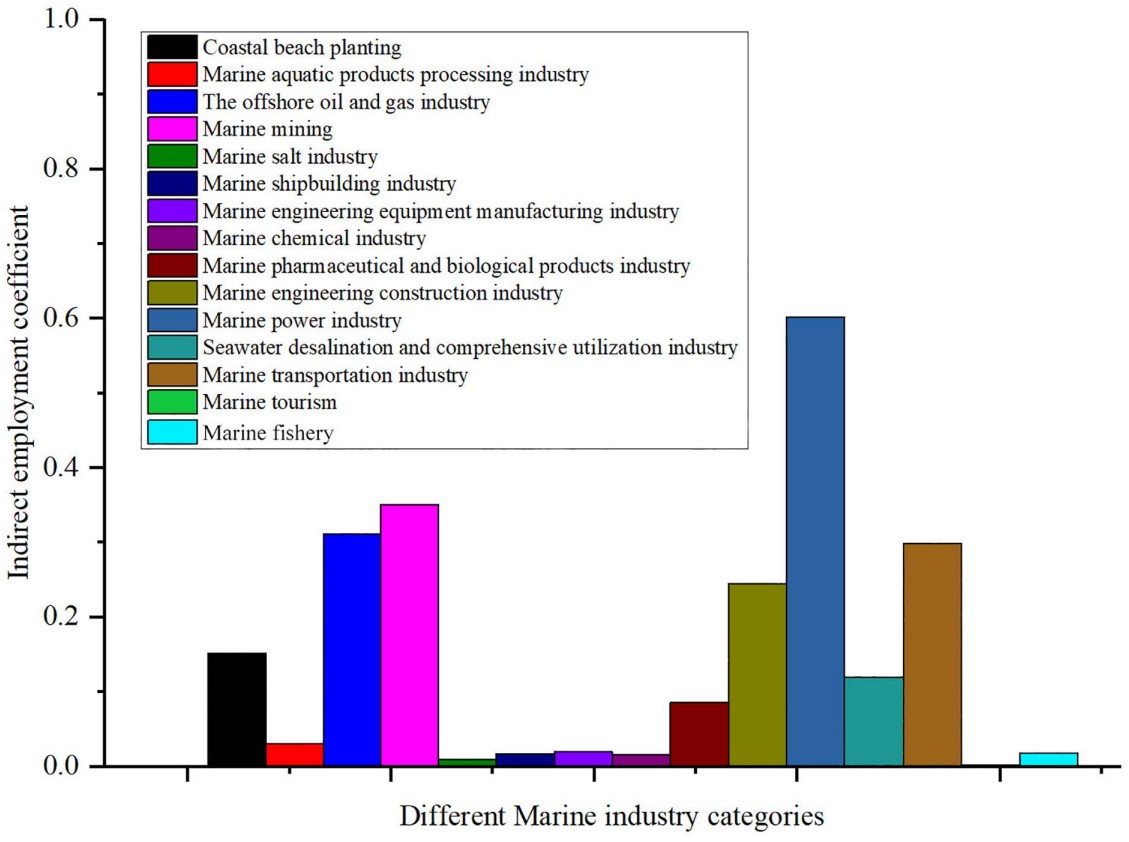

**Fig 5. Indirect employment coefficients for different marine industry categories.**

industry requires a large number of parts and materials, and these demands boost employment in related manufacturing and supply chains.

The indirect employment coefficients of the marine engineering equipment manufacturing industry and the marine chemical industry are 0.01954 and 0.01577, respectively, indicating that these two industries also have a certain contribution in driving employment. The marine engineering equipment manufacturing industry involves a large number of engineering design, technology development and equipment production, which require a variety of professionals and auxiliary services, thus indirectly driving the employment of related industries. The marine chemical industry, through its demand for chemical products, drives employment in related industries such as raw material supply and product sales.

The indirect employment coefficient of the coastal beach planting is 0.16351. It indirectly creates employment through the construction of seedling bases, the purchase of water quality monitoring equipment, and the processing of agricultural products. However, due to the low standardization of planting techniques, its supply chain synergy is weaker than that of marine mining. The indirect employment coefficient of the marine aquatic products processing industry is 0.02902. This industry mainly relies on cold chain transportation, food packaging equipment maintenance, and inspection and certification services. However, these links have limited employment-driven effects due to the improvement in automation levels. Currently, the proportion of deep processing in China is less than 20%, which restricts its potential to provide indirect employment.

The indirect employment coefficient of the offshore oil and gas industry is 0.31023. Oil and gas exploration and development drive the development of an industrial chain including high-end equipment manufacturing, technical services, and

port logistics. The technological intensity of this industry leads to the concentration of indirect employment in high-skilled fields, and the absorption capacity of ordinary labor is weaker than that of marine mining. The indirect employment coefficient of marine salt industry is 0.00998. The traditional salt-making process relies on simple tools and manual inspections. The indirect employment mainly focuses on the production of primary packaging materials and the supply of brine chemical raw materials. Due to the accelerated mechanization replacement, the positions for salt field maintenance have been decreasing year by year. The indirect employment coefficient of the marine engineering construction industry is 0.23989. It covers projects such as port construction and offshore wind power foundation construction. The indirect employment is distributed across the building materials supply chain, engineering supervision, and equipment leasing, etc. Engineering design positions with higher technical requirements account for a relatively large proportion.

The indirect employment coefficient of the seawater desalination and comprehensive utilization industry is 0.12894. It relies on high-tech links such as membrane material research and development, high-pressure pump manufacturing, and pipeline system installation, thereby driving specialized employment in the fields of materials science and precision equipment manufacturing. However, the core process automation rate is as high as 65% − 70%, and the demand for daily operation and maintenance positions is relatively limited. The indirect employment coefficient of the marine transportation industry is 0.30276. The expansion of ports has led to dredging projects, supervision services and logistics equipment leasing. The widespread adoption of automated terminals has resulted in a reduction of traditional loading and unloading positions.

The indirect employment coefficients of the marine engineering equipment manufacturing industry and the marine chemical industry are 0.01954 and 0.01577, respectively, indicating that these two industries also have a certain contribution in driving employment. The marine engineering equipment manufacturing industry involves a large number of engineering design, technology development and equipment production, which require a variety of professionals and auxiliary services, thus indirectly driving the employment of related industries. The marine chemical industry, through its demand for chemical products, drives employment in related industries such as raw material supply and product sales.

The indirect employment coefficient of the marine pharmaceutical and bioproducts industry is 0.08561, indicating that this industry has a great potential in driving employment in related industries. This may be due to the fact that the industry requires a lot of support for research and development, production and sales, which involves multiple links such as pharmaceutical research and development, production equipment, logistics and sales, thus indirectly creating a large number of employment opportunities. The indirect employment coefficient of the marine power industry is 0.60206, the highest among all industries, which shows the great potential of the marine power industry in driving employment in related industries. The marine power industry requires a large amount of equipment manufacturing, installation and maintenance, and these demands have significantly driven employment in machinery manufacturing, electronic components, installation engineering and other related industries.

Although the indirect employment coefficient of the marine tourism is only $4.6353E^{-5}$, which is insignificant, its direct employment coefficient is high, which means that it mainly creates employment through direct services and operations, rather than through indirect industrial chain. This is related to the characteristics of the tourism industry, whose employment is mainly concentrated in front-line service industries, such as hotels, catering and tour guides, rather than relying on complex supply chains and related industries.

## 5. Research on the multiplier effect of household income in China's marine economy

### 5.1 Multiplier effect of marine economy

The development of marine economy has a direct and indirect impact on the income of marine residents. The direct impact is mainly reflected in the fact that the increase of marine economic activities can directly provide employment opportunities, thus increasing the wage income of residents. For example, the successful implementation of marine service projects will help to create a large number of employment opportunities, including direct marine service

project-related jobs and indirect employment in ancillary industries, which will reduce the unemployment rate, improve the employment opportunities of residents and improve the quality of life.

Marine economy multiplier effect refers to the amplification effect of total output and income caused by the chain reaction triggered by the increase in investment and consumption in marine economic activities. This effect can be expressed in both direct and indirect aspects: the direct multiplier effect refers to the direct economic benefits generated by marine economic activities themselves, such as the direct output and income of the marine fishery, the marine tourism, the marine transportation industry and other industries. The indirect multiplier effect refers to the additional economic benefits generated by the expansion of marine economic activities, which drives the development of other related industries. For example, the development of the marine tourism may drive the growth of the hotel industry, catering industry, transportation industry and other related industries.

## 5.2 Structural path analysis

Structural path analysis of marine economy is a methodology used to study the development patterns and paths of marine economy [59]. Literature [33] identifies the path of high contribution rate in the Marine industry chain through SPA. This method usually involves the comprehensive consideration of the structure distribution, industrial development trend, resource utilization efficiency and environmental impact of marine economy. Through this analysis, the key factors in the development of marine economy can be identified, the future development trend can be predicted, and the scientific basis can be provided for the formulation of marine economic development strategy.

The modified Leontief inverse matrix is expanded by taylor series approximation as follows:

$$L^d = (I - A^d)^{-1} = I + A^d + (A^d)^2 + (A^d)^3 + \ldots + (A^d)^t \tag{17}$$

The equation is defined using the production layer $PL$ and a production layer ford each term in the power series expansion on the right: $(PL)^t = (A)^t$. Each additional layer, $(PL)^{t+1} = (PL)^t A$, represents that the output of intermediate products of the $t+1$ layer is the input of the $t$ layer, and the consumption of final demand $y$ can be defined as:

$$C = \varepsilon^d (I - A^d)^{-1} y^d = \varepsilon^d y^d + \varepsilon^d A^d y^d + \varepsilon^d (A^d)^2 y^d + \varepsilon^d (A^d)^3 y^d + \ldots + \varepsilon^d (A^d)^t y^d \tag{18}$$

where $\varepsilon^d (A^d)^t y^d$ represents the input-output of the $t$th production layer. For example, suppose that $y^d$ is the demand for a certain marine product, and $\varepsilon l y^d$ is the energy consumed by the marine industry company to produce this marine product. In order to produce marine products, the marine industry company needs to purchase from other industry sector investment $A^d y^d$ and the consumption $\varepsilon^d A^d y^d$ of these industry sectors. In turn, these industry sectors also require inputs (i.e. $A^d$ $(A^d y^d)$). And so on, the infinite expansion of the power series continues. Therefore, the main input in layer 0 is the input in the initial production stage of the energy product of the stage. The specific inputs of the first layer are the inputs related to the parts required by the marine industry company. The second or higher level of specific input use refers to the use of inputs produced for components in the supply chain. The number of nodes in a production network increases exponentially with each layer. Layer $t$ has $n^{t+1}$ nodes, in which $n$ is the number of industrial sectors in the economy.

## 5.3 Analysis of residents' income

Different sectors of China's marine economy play their own important roles in the marine income multiplier effect. The multiplier effect of China's marine income in 2023 is shown in Fig 6. Analyzing the marine revenue multiplier effect after removing individual sectors gives an idea of the extent to which each sector contributes to the overall economy. First of all, the income multiplier after removing the marine fishery is 5.998922872, which is slightly lower than the average of 6.203 for all sectors. This suggests that the marine fishery has a positive effect on income multiplier, but its importance is not the

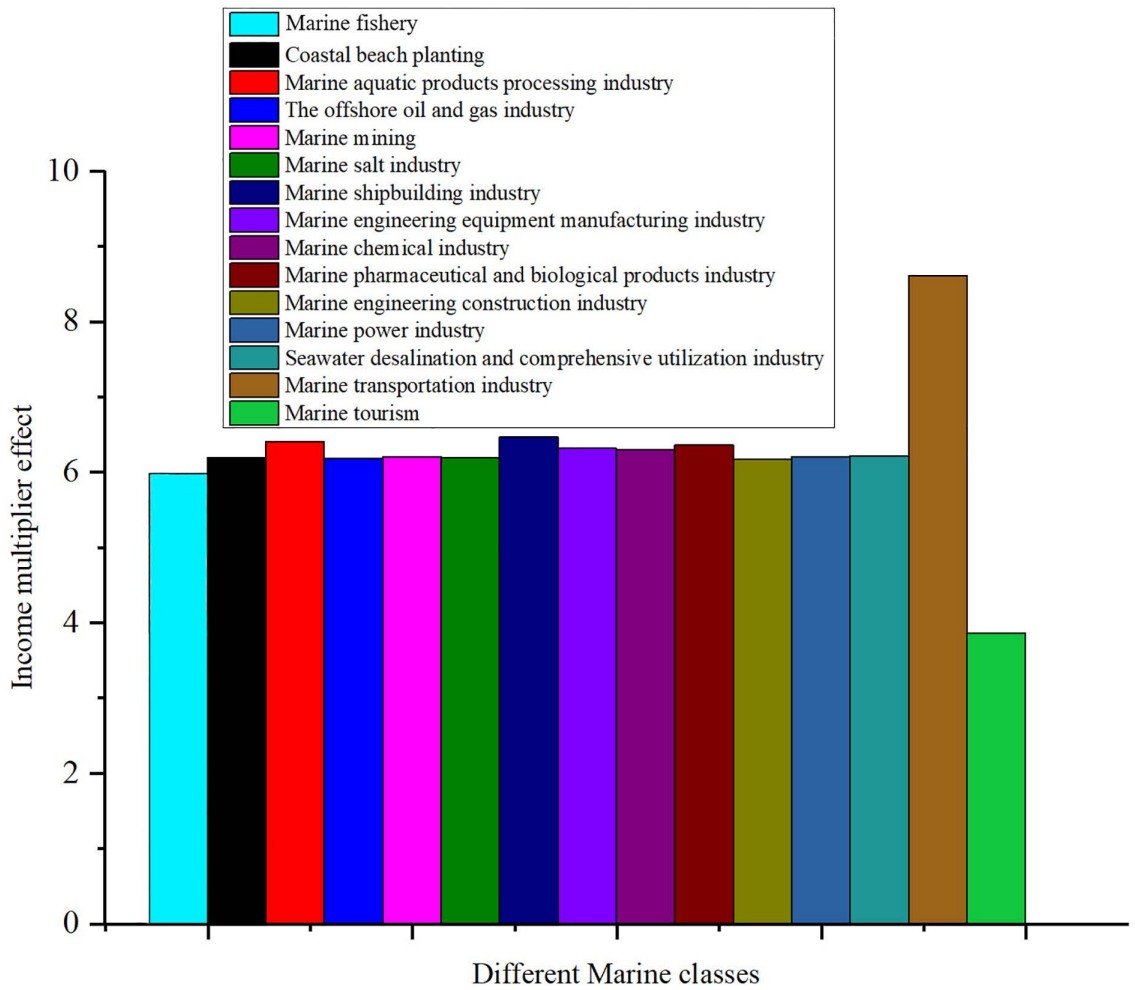

**Fig 6. Different marine industry income multiplier effect.**

highest relative to other departments. Similarly, the multiplier effect after the removal of the coastal beach planting industry and the marine aquatic products processing industry is 6.202759092 and 6.410921972, respectively, the former is almost consistent with the average, show that the minimal impact, while the latter is on the high side, it shows that the marine aquatic products processing industry occupies a more important position in the marine economy.

The marine oil and gas industry, marine mining, marine salt industry of the multiplier effect, 6.184969474, 6.205165499 and 6.20359796, respectively, with average close to show that these industries impact on marine income is relatively neutral, no significant increase or decrease the overall income multiplier. In contrast, the income multiplier after removing the marine shipbuilding industry is 6.469967312, which is significantly higher than the average, indicating that this industry has played a significant supporting role in the marine economy. A similar situation also appeared in the marine engineering equipment manufacturing industry and the marine chemical industry, after the removal of these two industries multiplier effect are 6.321814847 and 6.310262295, all show high influence.

In the marine pharmaceutical and biological products industry, the marine engineering construction industry, the marine power industry, after removing the multiplier effect were 6.36363963, 6.179579832 and 6.207492235 respectively. These Numbers show that the marine pharmaceutical and biological products industry has a larger positive effect on income

multiplier, and the influence of the marine engineering construction and the marine power industry is relatively moderate. It is worth noting that the multiplier effect of the marine desalination and comprehensive utilization industry is 6.217893706, which is close to the average value, indicating that its influence is relatively small.

When analyzing the impact of the marine transportation industry and the marine tourism, it is found that the multiplier effect of the marine transportation industry is significantly increased to 8.609299999 after the removal of the marine transportation industry, which is much higher than the average value, showing that the importance of this industry to the marine economy is irreplaceable. Instead, the marine tourism multiplier effect after removal only is 3.868774364, significantly lower than the average, show that the marine tourism has extremely significant positive contribution to the income multiplier, it had a huge impact.

The marine transportation industry has the largest impact on the marine income multiplier, and the multiplier effect after removing this industry increases by about 38.78% compared to the average. This shows that the marine transportation industry plays a key role in driving China's marine economy, and its importance far exceeds that of other industries. As an advanced form of the Marine transportation industry, the policy design of free trade port can significantly amplify the income multiplier effect. The policy innovation of Hainan Free Trade Port in China provides a new practical case. Through the design of "zero tariff, low tax rate and simple tax system", Hainan Free Trade Port has greatly reduced the institutional transaction cost. In particular, its financial opening policy, which allows the free flow of funds across borders, has promoted the development of offshore financial business [60]. Empirical studies in the Yangtze River Delta region show that after the integrated development of tourism through high-speed rail network, the regional average tourism revenue multiplier increases, which is mainly due to the sharing of tourist sources and the integration of industrial chain brought by transportation interconnection [61].

China's marine employment multiplier effect distribution is shown in Fig 7. Each sector of China's marine economy contributes differently to the marine employment multiplier effect, and the impact of each industry on overall employment can be clearly seen through data analysis after removing individual sectors. First of all, the employment multiplier effect after the marine fishery is removed is 2.989430357, which is significantly higher than the overall average of 2.8, indicating that the marine fishery plays an important role in promoting employment, and the removal of this sector will significantly increase the overall employment multiplier. The employment multiplier effect of the coastal beach planting industry and the marine aquatic products processing industry after removal is 2.796060572 and 2.733442489 respectively, both lower than the average, especially the employment multiplier effect of marine aquatic product processing industry is close to the average, indicating that these industries have a small impact on employment, and the effect of removing them on the overall employment multiplier is not significant.

Further analysis shows that the employment multiplier effects of the marine oil and gas industry, the marine mining and the marine salt industry after removal are 2.7697546, 2.791052482 and 2.79386581 respectively, which are close to the average value of 2.8. These industries have a relatively neutral effect on the marine employment multiplier, and their removal does not change the overall employment multiplier much. In contrast, the removal of the marine shipbuilding industry and the marine engineering equipment manufacturing industry makes the employment multiplier effect 2.729450498 and 2.759788226, respectively, in particular, the value of the marine shipbuilding industry is significantly lower than the average, showing the low importance of this industry for overall employment.

The employment multiplier effects of the marine chemical industry and the marine pharmaceutical and biological products industry after removal are 2.769327662 and 2.776867013 respectively, which are slightly lower than the average, indicating that although the contribution of these industries in employment exists, it is not prominent. The removal effects of the marine engineering construction industry and the marine power industry are 2.754707631 and 2.791912323, respectively, slightly higher than the overall average, indicating that these industries have a certain role in promoting employment, but the impact is not particularly significant. Similarly, the employment multiplier effect of the marine desalination and comprehensive utilization industry after removal is 2.793452135, which is close to the average value, indicating that it has little impact on overall employment.

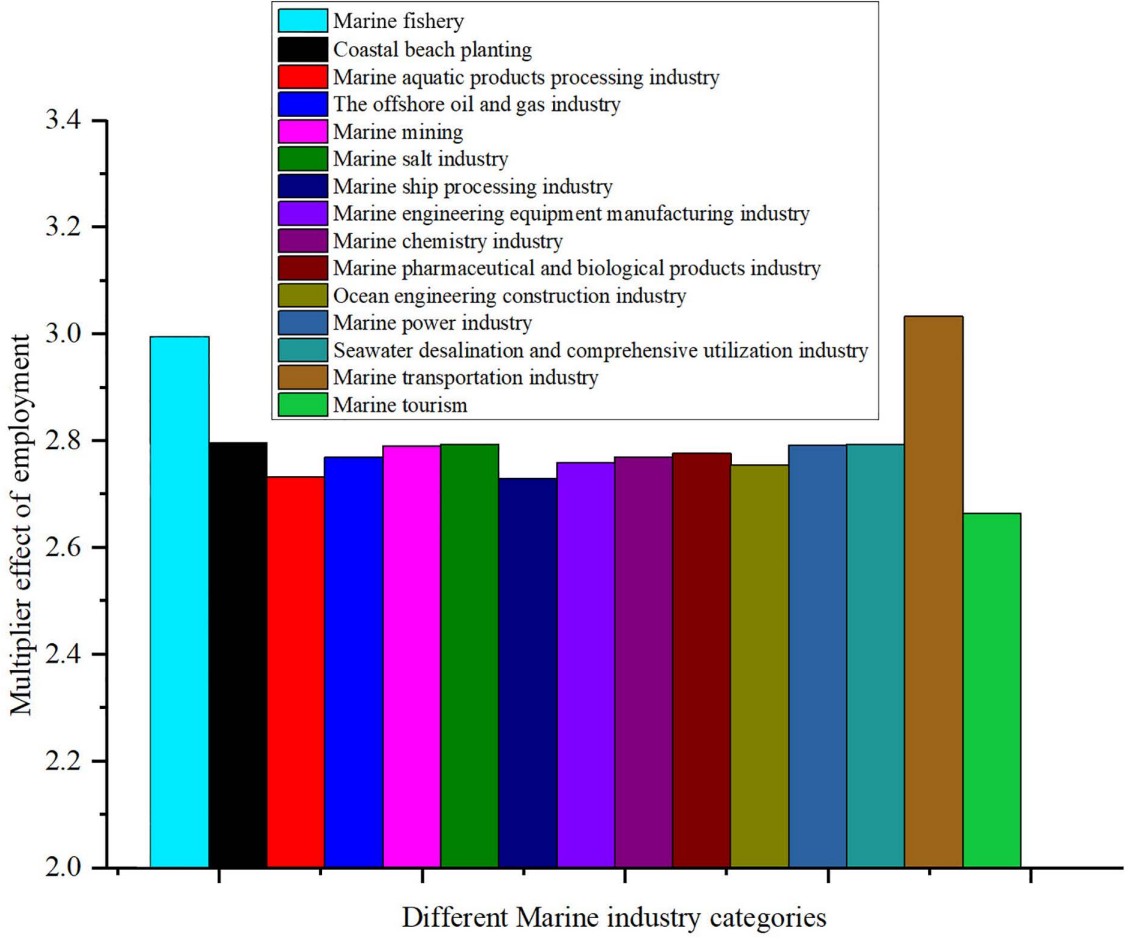

**Fig 7. Different marine industry multiplier effect of employment.**

When analyzing the impact of the marine transportation industry and the marine tourism on the employment multiplier, it is found that the employment multiplier effect after removing the marine transportation industry is 3.033204803, which is significantly higher than the average, indicating the important contribution of this industry to the overall employment. After the removal of the marine tourism, by contrast, employment multiplier is 2.664619309, is lower than the average, show that the marine tourism has very important role in promoting employment, its influence is most significant in all departments.

The marine tourism has the largest impact on the marine employment multiplier, and the employment multiplier effect after removing this industry is reduced by about 4.84% compared to the overall average. This shows that the marine tourism plays a key role in promoting China's marine economy, and its importance is significantly higher than that of other industries.

## 6. Policy recommendations

Achieving balanced development of marine industry is the key to improving the quality of marine economy. In view of the current structural contradictions in China's marine industry, it is necessary to build a systematic and precise policy system to promote the upgrading of traditional industries, the cultivation of emerging industries and the coordinated development

of industries. Based on the latest research results, this part puts forward specific policy suggestions to promote the balanced development of Marine industry.

The transformation and upgrading policies of traditional marine industries should focus on improving the quality and efficiency of fishery, shipping and other industries. For marine fisheries, it is necessary to strictly implement the targets of the 14th Five-Year Plan and control the intensity of offshore fishing [62]. At the same time, we will increase policy support for pelagic fisheries, improve the industrial chain of pelagic fisheries, and encourage enterprises to update equipment and expand their operating areas through financial subsidies and tax incentives. For the Marine transportation industry, it is suggested to implement a subsidy policy for green port construction, promote the electric transformation of port operation equipment, and give priority to berthing and port fee reduction and exemption to ships that use clean energy.

The cultivation policy of strategic emerging industries needs to take precise measures according to the characteristics of different industries. In the field of marine biomedicine, it is proposed to set up a national special fund for marine biomedicine research and development, support industry-university-research collaboration to tackle key problems, and provide rapid approval and exclusive market period policies for marine innovative drugs [63]. In the field of offshore power, we will improve the feed-in tariff subsidy mechanism for offshore wind power, implement differentiated subsidy policies, and focus on supporting the research and development and demonstration application of far-reaching offshore wind power technologies. For the marine equipment manufacturing industry, we will implement an insurance compensation mechanism for the first equipment to reduce users' use risks and create market demand through government procurement.

The policy of coordinated industrial development focuses on building a modern Marine industrial system. It is suggested to compile a guidance catalogue for the integrated development of Marine industries, and specify the integrated business forms and key areas to encourage development. We will support the establishment of Marine industry innovation alliances to promote technology and business integration between traditional and emerging Marine industries. For example, we will promote the integration of Marine fisheries and tourism, and develop new forms of business such as recreational sea fishing and fishing. In terms of financial support, banks are encouraged to set up special loans for Marine economy, develop "Marine credit" products, and provide discount interest support to qualified Marine industry integration projects [64]. We will support qualified Marine enterprises in issuing green bonds, and the funds raised will be specifically used for industrial upgrading and integrated development projects.

## 7. Conclusions

This paper mainly uses Ghosh model, employment model, structural path analysis to analyze output effect, employment effect and resident income multiplier effect of China's marine economy, the integration of neural networks with the Ghosh model significantly improves the accuracy and depth of marine economic analysis, integrating neural networks with the Ghosh model, thereby overcoming the limitations of traditional methods in terms of dynamics, nonlinearity and robustness. This interdisciplinary approach enhances the accuracy of predicting the marine economic multiplier effect, initiates a "data-driven - policy iteration" sustainable governance framework, and encourages intelligent analysis of global resource economies.

In terms of the leading contribution of China's marine industry, marine tourism is the core contributor of value added (44.346%) and total output (48.87%), highlighting its position as a pillar industry. The direct employment coefficient of marine fishery is the highest (0.42681), providing the most employment opportunities, while the Marine power industry significantly drives the employment growth of related industries through the indirect employment driving effect (coefficient: 0.60206). The marine transportation sector shows the most critical driving role in both the income multiplier and the employment multiplier.

This study has the following limitations that should be treated with caution:

1) Data dependence and timeliness: some industry data rely on stripping coefficient estimation, which may introduce bias; The lack of completeness of Marine economic statistical data affects the comprehensiveness of the model.

2) Model complexity and applicability: although LSTM model can capture nonlinear relationships, training of high-dimensional data requires a lot of computing resources and has the risk of overfitting, which may be restricted by hardware conditions in practical application.

3) Lack of demand-side and ecological factors: focus on supply-side dynamics, but demand-side variables are not fully integrated, which may underestimate the comprehensive impact of external shocks; Ecological costs have not yet been quantified, making it difficult to fully assess long-term sustainability.

Based on the above limitations, future research can be further developed from the following directions:

1) Data and model optimization: integrating high-frequency micro enterprise data and environmental indicators to construct an environmentally extended input-output table; Try to integrate Transformer or graph neural network to improve the modeling efficiency of cross-department spatio-temporal dependence.

2) Multi-dimensional dynamic framework: Leontief demand-pull model and Ghosh supply-side model are combined to establish a three-dimensional analysis framework of "supply-demand-ecology" to simulate the synergistic effect of policy and market and the resilient response to extreme events.

3) Emerging fields and policy verification: deepen the technology-economic path analysis of strategic emerging industries such as marine biomedicine and marine power; Develop a policy experiment platform to quantify the role of blue financial instruments in promoting low-carbon transformation.

The dynamic model of this study provides actionable tools for policy makers: by optimizing resource allocation and upgrading infrastructure, the marine economy can be transformed into high-quality, low-carbon, and inclusive development, contributing to the realization of China's strategic goal of becoming a maritime power.

## Supporting information

**S1 Table. Input-output table forecasting.**
(XLSX)

**S2 Table. Classification of marine industries and the corresponding relationship between China's input-output table in 2020 (153 sectors).**
(XLSX)

## Author contributions

**Conceptualization:** Jian Jin, Mingqi Zhang.

**Data curation:** Mingqi Zhang.

**Formal analysis:** Mingqi Zhang.

**Investigation:** Mingqi Zhang.

**Methodology:** Mingqi Zhang.

**Project administration:** Jian Jin.

**Resources:** Jian Jin.

**Software:** Mingqi Zhang.

**Supervision:** Jian Jin.

**Validation:** Jian Jin, Mingqi Zhang.

**Visualization:** Jian Jin.

**Writing – original draft:** Mingqi Zhang.

**Writing – review & editing:** Mingqi Zhang.

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
