## [Decision Letter · Decision Letter 0]

26 Jun 2025

Dear Dr. Zhang,

Thank you for submitting your manuscript to PLOS ONE. After careful consideration, we feel that it has merit but does not fully meet PLOS ONE’s publication criteria as it currently stands. Therefore, we invite you to submit a revised version of the manuscript that addresses the points raised during the review process.

We look forward to receiving your revised manuscript.

Kind regards,

Tien Anh Tran

Academic Editor

PLOS ONE

Journal Requirements:

Additional Editor Comments:

**Thank you so much for your paper. The paper must be addressed all comments of editors and reviewers for improving the quality of research. There are some additional comments that the authors should address them to improve the paper.**
**1. In abstract, the certain results of this study must be provided clearly under the quantitative values to evaluate the marine business and its related policy.**
**2. There are some statements that the authors referred into main text: In this paper, we integrate neural networks with the Ghosh model to address limitations in capturing dynamic interdependencies and improve the robustness of marine economies analysis." The authors must make clear the significance and novelty points from this proposal.**
**3. The research gap should be provided in the introduction.**

Reviewers' comments:

Reviewer's Responses to Questions

**Comments to the Author**

1. Is the manuscript technically sound, and do the data support the conclusions?

Reviewer #1: Partly

Reviewer #2: Partly

2. Has the statistical analysis been performed appropriately and rigorously?

Reviewer #1: Yes

Reviewer #2: N/A

3. Have the authors made all data underlying the findings in their manuscript fully available?

Reviewer #1: No

Reviewer #2: Yes

4. Is the manuscript presented in an intelligible fashion and written in standard English?

Reviewer #1: Yes

Reviewer #2: No

Reviewer #1: The abstract needed to be revised as follows, objective, method, results, and contribution.

The past studies are not up-to-date and comprehensive enough needed to be updated with the most recent papers related to the models used.

The manuscript has not addressed the research gaps in past studies to fill them and show the study contributions in the models used.

The establishment of the models and showing their contribution to the knowledge gaps based on the past studies gaps filled to show the study contributions should be written.

The results generated should be explained based on the study issues and the findings of past studies should be written based on the comparison of past studies' findings.

It should be recommendations based on the study findings and policy implications based on the study’s most significant findings.

The study limitations and suggestions for further studies should be added.

Reviewer #2: This article takes China's marine economy as the research object and employs the Ghosh input-output model, employment model, and structural path analysis, among other methods, to conduct an in-depth analysis of the output effect, employment effect, and resident income multiplier effect of China's marine economy from 2017 to 2023. It also enhances the prediction accuracy by integrating neural network technology, providing a valuable reference for the sustainable development of the marine economy and holding certain research value. However, the paper has significant deficiencies in terms of methodological rigor, data transparency, depth of result interpretation, and writing standardization, and thus requires substantial revisions before it can be considered for acceptance.

1. The abstract can further emphasize the innovative points of the research, such as the application of the combination of neural networks and the Ghosh model in the field of marine economy, etc., to highlight its contributions.

2. When introducing the research question, it is advisable to more specifically point out the existing deficiencies in the current research on the sustainable development of the marine economy, especially by integrating the global context and the progress of international research, so as to better highlight the necessity and innovation of this study.

3. The selection and calculation methods of key data, such as value-added rate and stripping coefficient, should be supplemented with more detailed references or case studies to enhance the credibility of the data.

4. The method section can further compare and analyze the advantages and disadvantages of the Ghosh model and the Leontief model in marine economic research, and more deeply explore their applicability and limitations.

5. In the analytical process, more practical cases or policy backgrounds can be incorporated to further explore the interrelationships among various marine industries and the mechanisms through which they influence the sustainable development of the marine economy. Currently, most analyses remain at the level of presenting results, lacking in-depth analysis.

6. Discussion is of great significance and serves as the elevation of the article. Currently, the discussion is not clear. It is suggested to add a discussion section to compare and analyze the reliability and innovativeness of the research results with existing studies, and to point out the limitations of this research.

7. More targeted policy suggestions can be put forward, such as how to achieve balanced development of various marine industries through policy guidance and how to enhance the coordination between marine environmental protection and economic development.

**Do you want your identity to be public for this peer review?** For information about this choice, including consent withdrawal, please see our Privacy Policy

Reviewer #1: **Yes: ** Elsadig Musa Ahmed

Reviewer #2: No

---

## [Author Response · Author response to Decision Letter 1]

21 Jul 2025

Dear Editors and Reviewers,

We sincerely appreciate the editor' and reviewers' constructive feedback and have carefully addressed each comment to improve the manuscript. Below is our point-by-point response:

Editor

1. In abstract, the certain results of this study must be provided clearly under the quantitative values to evaluate the marine business and its related policy.

As suggested, we have now included key quantitative findings in the manuscipt, ensuring readers can immediately evaluate the implications for marine economy and policy.

2. There are some statements that the authors referred into main text: In this paper, we integrate neural networks with the Ghosh model to address limitations in capturing dynamic interdependencies and improve the robustness of marine economies analysis." The authors must make clear the significance and novelty points from this proposal.

We have expanded the manuscipt to explicitly state the novelty and significance of our approach.

3. The research gap should be provided in the introduction.

We have restructured the introduction to clearly delineate the research gap.

Reviewer #1

1. The abstract needed to be revised as follows, objective, method, results, and contribution.

We have restructured the abstract to explicitly include four key subsections: objective, methods, results, and contributions, ensuring alignment with journal guidelines and enhancing clarity.

2. The past studies are not up-to-date and comprehensive enough needed to be updated with the most recent papers related to the models used.

We have extensively updated the literature review, adding a lot of recent publications focusing on the models applied in our study, ensuring a state-of-the-art foundation.

3. The manuscript has not addressed the research gaps in past studies to fill them and show the study contributions in the models used.

We have added a dedicated paragraph explicitly outlining unresolved limitations in prior work and how our study bridges these gaps.

4. The establishment of the models and showing their contribution to the knowledge gaps based on the past studies gaps filled to show the study contributions should be written.

We now explicitly detail how each proposed model addresses specific limitations identified in prior literature, thereby demonstrating their novel contributions to advancing knowledge.

5. The results generated should be explained based on the study issues and the findings of past studies should be written based on the comparison of past studies' findings.

We have expanded interpretation of key results in the context of research questions.

6. It should be recommendations based on the study findings and policy implications based on the study’s most significant findings.

We translate key findings into actionable recommendations for practitioners and policymakers.

7. The study limitations and suggestions for further studies should be added.

We have incorporated limitations and future research, discussing constraints.

Reviewer #2

1. The abstract can further emphasize the innovative points of the research, such as the application of the combination of neural networks and the Ghosh model in the field of marine economy, etc., to highlight its contributions.

We have significantly revised the abstract to explicitly underscore the novel integration of neural networks with the Ghosh input-output model as a core contribution. We now clearly articulate how this hybrid approach uniquely addresses dynamic and nonlinear interdependencies within marine economic systems, and explicitly state its pioneering application in marine sustainability research.

2. When introducing the research question, it is advisable to more specifically point out the existing deficiencies in the current research on the sustainable development of the marine economy, especially by integrating the global context and the progress of international research, so as to better highlight the necessity and innovation of this study.

Section 1 (Introduction) has been comprehensively revised to incorporate recent international studies, explicitly identifying three critical limitations in current marine sustainability literature. This contextualization solidifies the necessity and originality of our approach.

3. The selection and calculation methods of key data, such as value-added rate and stripping coefficient, should be supplemented with more detailed references or case studies to enhance the credibility of the data.

We have supplemented the manuscipt with more detailed references to enhance the credibility of the data.

4. The method section can further compare and analyze the advantages and disadvantages of the Ghosh model and the Leontief model in marine economic research, and more deeply explore their applicability and limitations.

Section 3.3 now includes a systematic comparison contrasting the Ghosh and Leontief models. We critically discuss why the Ghosh model’s supply-side perspective offers a distinct advantage for analyzing resource-driven marine economies.

5. In the analytical process, more practical cases or policy backgrounds can be incorporated to further explore the interrelationships among various marine industries and the mechanisms through which they influence the sustainable development of the marine economy. Currently, most analyses remain at the level of presenting results, lacking in-depth analysis.

We have incorporated practical case studies and policy contexts into the revised manuscript, providing a deeper analysis of the interrelationships among marine industries and their impact mechanisms on sustainable development. These enhancements strengthen the practical relevance and analytical rigor of our study.

6. Discussion is of great significance and serves as the elevation of the article. Currently, the discussion is not clear. It is suggested to add a discussion section to compare and analyze the reliability and innovativeness of the research results with existing studies, and to point out the limitations of this research.

We have now added a detailed discussion that thoroughly compares the reliability and innovativeness of our results with prior studies, while also explicitly addressing the limitations of our current research.

7. More targeted policy suggestions can be put forward, such as how to achieve balanced development of various marine industries through policy guidance and how to enhance the coordination between marine environmental protection and economic development.

We have carefully addressed the reviewer's suggestions by refining our policy recommendations to promote balanced marine industry development and improved coordination between environmental protection and economic growth, as detailed in the revised manuscript.

We believe these revisions address all concerns and significantly enhance the paper’s rigor, clarity, and impact. Thank you for your valuable guidance.

Sincerely,

Mingqi Zhang

---

## [Decision Letter · Decision Letter 1]

26 Sep 2025

Dynamic supply-side multipliers in China's marine economy: a neural network-enhanced Ghosh model for sustainable development

PONE-D-25-21847R1

Dear Dr. Zhang,

We’re pleased to inform you that your manuscript has been judged scientifically suitable for publication and will be formally accepted for publication once it meets all outstanding technical requirements.

Kind regards,

Tien Anh Tran

Academic Editor

PLOS ONE

Additional Editor Comments (optional):

Reviewers' comments:

Reviewer's Responses to Questions

**Comments to the Author**

Reviewer #1: All comments have been addressed

2. Is the manuscript technically sound, and do the data support the conclusions?

Reviewer #1: Yes

3. Has the statistical analysis been performed appropriately and rigorously?

Reviewer #1: Yes

4. Have the authors made all data underlying the findings in their manuscript fully available?

Reviewer #1: Yes

5. Is the manuscript presented in an intelligible fashion and written in standard English?

Reviewer #1: Yes

Reviewer #1: The comments applied applied to a great extent, the manuscript should be check carefully by the authors and the editor.

**Do you want your identity to be public for this peer review?** For information about this choice, including consent withdrawal, please see our Privacy Policy

Reviewer #1: **Yes: ** Elsadig Musa Ahmed

---

## [Editor Report · Acceptance letter]

PONE-D-25-21847R1

PLOS ONE

Dear Dr. Zhang,

I'm pleased to inform you that your manuscript has been deemed suitable for publication in PLOS ONE. Congratulations! Your manuscript is now being handed over to our production team.

Kind regards,

on behalf of

Professor Tien Anh Tran

Academic Editor

PLOS ONE